# The Effect of Green and Black Tea Polyphenols on *BRCA2* Deficient Chinese Hamster Cells by Synthetic Lethality through PARP Inhibition

**DOI:** 10.3390/ijms20061274

**Published:** 2019-03-14

**Authors:** Shaherah Alqahtani, Kelly Welton, Jeffrey P. Gius, Suad Elmegerhi, Takamitsu A. Kato

**Affiliations:** 1Department of Environmental & Radiological Health Sciences, Colorado State University, Fort Collins, CO 80523, USA; Shaherahtoxicology@gmail.com (S.A.); kelly.welton@hotmail.com (K.W.); Jeff.gius8@gmail.com (J.P.G.); Hanoyara@rams.colostate.edu (S.E.); 2Cell Molecular Biology Program, Colorado State University, Fort Collins, CO 80523, USA

**Keywords:** theaflavin, epigallocatechin gallate, PARP, BRCA2, synthetic lethality

## Abstract

Tea polyphenols are known antioxidants presenting health benefits due to their observed cellular activities. In this study, two tea polyphenols, epigallocatechin gallate, which is common in green tea, and theaflavin, which is common in black tea, were investigated for their PARP inhibitory activity and selective cytotoxicity to *BRCA2* mutated cells. The observed cytotoxicity of these polyphenols to *BRCA2* deficient cells is believed to be a result of PARP inhibition induced synthetic lethality. Chinese hamster V79 cells and their *BRCA2* deficient mutant V-C8, and V-C8 with gene complemented cells were tested against epigallocatechin gallate and theaflavin. In addition, Chinese hamster ovary (CHO) wild-type cells and rad51D mutant 51D1 cells were used to further investigate the synthetic lethality of these molecules. The suspected PARP inhibitory activity of epigallocatechin and theaflavin was confirmed through in vitro and in vivo experiments. Epigallocatechin gallate showed a two-fold increase of cytotoxicity to V-C8 cells compared to V79 and gene complimented cells. Compared to CHO wild type cells, 51D1 cells also showed elevated cytotoxicity following treatment with epigallocatechin gallate. Theaflavin, however, showed a similar increase of cytotoxicity to VC8 compared to V79 and gene corrected cells, but did not show elevation of cytotoxicity towards rad51D mutant cells compared to CHO cells. Elevation of sister chromatid exchange formation was observed in both tea polyphenol treatments. Polyphenol treatment induced more micronuclei formation in *BRCA2* deficient cells and *rad51D* deficient cells when compared against the respective wild type cells. In conclusion, tea polyphenols, epigallocatechin gallate, and theaflavin may present selective cytotoxicity to *BRCA2* deficient cells through synthetic lethality induced by PARP inhibition.

## 1. Introduction

Tea is the most widely consumed beverage in the world, after water, and it is sold in three major forms: green, oolong, and black. All three teas come from the same plant, *Camellia sinensis*, and the only difference between the three types of tea is the level of oxidation polyphenols, a major class of molecules in tea leaves, by fermentation. Green tea is prevented from oxidizing at all, oolong is only partially oxidized, and black tea polyphenols are entirely oxidized in a process known as fermentation. Catechins, members of the flavanol family, are heavily prevalent in green teas and are excellent antioxidants that have been observed to scavenge free radicals and reduce oxidative stress [1]. The concentration of polyphenols in Green tea are measured to be 200 to 300 mg/cup while concentrations found in black tea are ~16–24 mg/cup [2,3]. Epigallocatechin gallate (Figure 1A) is an unoxidized polyphenol that is considered one of the most potent dietary antioxidants within green tea [4]. When black tea is manufactured, the catechin undergoes oxidation into quinone, which is further condensed into several other chemicals, one of which is theaflavin (Figure 1B). Theaflavin is responsible for the characteristic color and taste of black tea, and theaflavin levels are directly correlated with the quality and taste of the tea [1]. The health effects of both types of tea have been previously studied; one study showed that daily consumption of green tea decreased the number of lymph node metastases and decreased the reoccurrence of stage I and stage II breast cancers [5]. A comprehensive meta-analysis confirmed that consumption of green tea significantly lowered the risk of breast cancer while results for black tea consumption were mixed [6]. It is possible that some of the beneficial effects of green and black tea are due to the high concentrations of antioxidants in the form of flavonoids. Polyphenols have long been known to possess many health benefits. Polyphenols have a low molecular weight, are phenolic, and are found in most fruits, vegetables, nuts, herbs, and teas. The major benefits of polyphenols have been most documented in the treatment or prevention of adverse health effects and diseases such as diabetes, inflammation, and cardiovascular disease. They have also been shown to exhibit antiproliferation, apoptotic, and angiogenesis inhibiting effects in cancer cells, resulting in an observable level of antitumor capabilities [7,8]. Moreover, tea polyphenols have also been reported for their protective action against ionizing radiation by their antioxidant capacity [9].

It has been observed that many polyphenolic flavonoids can induce PARP inhibition [10,11]. PARP proteins are essential for the DNA repair process known as base excision repair (BER) and have strong downstream effects when inhibited [12]. One downstream effect of PARP inhibition is observed by selective cytotoxic effects. Specifically, it is observed that PARP inhibition is selectively lethal towards *BRCA2* deficient cell cultures and tumors. BRCA2, like PARP, is an important protein in DNA repair. However, unlike PARP, BRCA2 is primarily involved in the repair of double stranded DNA lesions through a pathway known as homologous recombination (HR) repair. HR repair is mediated by many proteins including BRCA1, BRCA2, and rad51D. Inhibition or mutation of any of these proteins can result in the inaccessibility of the HR pathway by cells to repair double stranded damage. When this occurs, cells are forced to utilize other more error prone and dangerous pathways, such as non-homologous end joining (NHEJ) repair. Due to the essential functions of both PARP and BRCA2, the loss of activity of both simultaneously can result in cellular death through a process known as synthetic lethality [13]. Synthetic lethality is a result of an accumulation of single strand DNA breaks, which if not corrected through BER, can result in the subsequent formation of double stranded DNA breaks through replication machinery failure. Repair of double stranded DNA breaks through pathways like NHEJ can cause further mutation and can result in cell death. Cancers with BRCA2 homozygous mutations have been proven to be very sensitive to treatment with PARP inhibitors like olaparib [14].

The objective of this study was to determine which polyphenols in tea, epigallocatechin gallate or theaflavin, contained the highest level of selective cytotoxicity towards *BRCA2* deficient cells through inhibition of PARP. In order to test our hypothesis, Chinese hamster V79 cells, their *BRCA2* deficient mutant V-C8 cells, and V-C8 gene complimented cells were utilized along with Chinese hamster ovary (CHO) cells and *rad51D* mutated 51D1 cells.

## 2. Results

### 2.1. Clonogenic Cell Survival

To determine if these polyphenols influence survival of *BRCA2* deficient cells, Chinese hamster lung origin cells were treated with various concentrations of each polyphenol and were incubated until colonies were formed. Treatment of cells by epigallocatechin gallate strongly suppressed clonogenic activity for *BRCA2* deficient V-C8 cells compared to wild type V79 cells and gene complimented cells (Figure 2A,B). The IC_50_ values were 57.1, 55.6, and 29.9 µM for V79, gene complimented cells, and V-C8 cells, respectively. The survival fraction at 50 µM showed statistically significant difference for V-C8 cells compared to V79 and gene complimented cells (*p* < 0.05). Similarly, treatment of cells by theaflavin also strongly suppressed clonogenic activity for *BRCA2* deficient V-C8 cells compared to wild type V79 cells and gene complimented cells. The IC_50_ values were 79.7, 80.0, and 54.3 µM for V79, gene complimented cells, and V-C8 cells, respectively. The survival fraction at 100 µM showed statistically significant difference for V-C8 cells compared to V79 and gene corrected cells (*p* < 0.05). Therefore, both epigallocatechin gallate and theaflavin presented selective cytotoxicity toward to *BRCA2* deficient cells (Figure 2C,D).

In order to expand this finding to other homologous recombination repair deficient cells, CHO wild type cells and *rad51D* mutated 51D1 cells were utilized. Rad51 and BRCA2, as described earlier, are essential for HR repair function. Therefore, if both polyphenols showed selective cytotoxicity to 51D1 cells, the polyphenol effects can be expanded to all HR repair defective cells. Treatment of cells by epigallocatechin gallate suppressed clonogenic activity for *rad51d* deficient 51D1 cells compared to wild type CHO cells. The IC_50_ values were 50.2 and 40.1 µM for CHO and 51D1 cells, respectively. The survival fraction at 50 µM showed statistically significant difference for 51D1 cells compared to CHO10B2 cells (*p* < 0.05). However, treatment of cells by theaflavin did not suppress clonogenic activity for 51D1 cells compared to wild type CHO cells. The IC_50_ values were 50.0 and 54.2 for CHO and 51D1 cells, respectively. The survival fraction did not show a statistically significant difference between CHO wild type and 51D1 cells for theaflavin treatment. Therefore, tea polyphenol epigallocatechin gallate may present selective cytotoxicity to other homologous recombination deficient cells but theaflavin may be more limited to *BRCA2* deficient cells.

### 2.2. Growth Delay and Cell Cycle Analysis

Growth delay by tea polyphenols was tested by adding 10, 30, and 50 µM of the polyphenols to cell cultures. The number of cells was recorded daily and population doubling hours were obtained (Figure 3). For V79 cells, treatment of cells with both epigallocatechin gallate and theaflavin showed polyphenol concentration dependent growth delay. The basal doubling time was 12.9 h for V79 and 17.4 h for V-C8. No delay or a minor delay was observed for 10 µM of epigallocatechin gallate treatment and 10 and 30 µM of theaflavin for V79 system cell lines. However, 50 µM of epigallocatechin gallate or theaflavin, particularly 50 µM of epigallocatechin gallate, strongly suppressed cellular growth. There was no clear cell line dependency in this aspect of the study. Statistically significant cell doubling time delay was observed for 50 µM of epigallocatechin gallate treated V79 and V-C8 cells (both *p* < 0.05).

For CHO and 51D1 cells, the effect of polyphenol for cell doubling time was much smaller than for the V79 cell system. A concentration dependent cellular growth delay was observed for both polyphenols and cell types. The basal doubling time was 12.1 h for CHO and 16.4 h for 51D1. The 50 µM of epigallocatechin and theaflavin increased CHO doubling time to 15.5 h and 15.2 h, respectively, and increased 51D1 doubling time to 22.1 h and 19.2 h, respectively. DNA repair deficiency specific growth delay was not observed for tested conditions.

### 2.3. In Vitro and In Vivo PARP Activity Inhibition

In vitro polyphenol effects for PARP activity were analyzed with in vitro PARP activity kits (Figure 4A). Quercetin was used as positive control. As predicted from their chemical structures, it was identified that epigallocatechin gallate and theaflavin showed PARP inhibitory effects and significant inhibition at 10 µM (*p* < 0.05). Statistically, theaflavin showed the stronger reduction of PARP activity (0.20, *p* < 0.0001) than epigallocatechin gallate (0.45, *p* < 0.0002). PARP inhibition effects by tea polyphenols were comparable to quercetin, which was formally reported for the natural flavonoids with strong PARP inhibition [10].

In vivo PARP inhibitory effect was tested with V79 cells. Prior to H_2_O_2_ treatment, 10 µM of epigallocatechin gallate or theaflavin were added to cell culture media, and their PARP inhibitory effects were assessed by measuring poly (ADP-ribose) formation in cells (Figure 4B). H_2_O_2_ treatment increased immunofluorescent cellular poly (ADP-ribose) signal from 1 to 9. Epigallocatechin gallate or theaflavin pretreatment suppressed H_2_O_2_ induced poly (ADP-ribose) formation (3 and 2, respectively).

### 2.4. Sister Chromatid Exchanges

A sister chromatid exchange (SCE) assay was performed to determine if these polyphenols increase the genetic instability to the cells as a function of the PARP inhibitor or other mechanisms (Figure 5A). The background SCE frequency was 6.3 SCE per CHO metaphase cell. Treatment of 10, 30, and 50 µM of tea polyphenol increased SCE frequency in a dose dependent manner (Figure 5B). The regression lines showed SCE frequency increases 0.061 per µM of epigallocatechin treatment and 0.038 per µM of theaflavin treatment. The rate of increase was greater for epigallocatechin treatment. The statistically significant increase was observed for 30 and 50 µM of epigallocatechin and 50 µM of theaflavin treatment against control. Therefore, both epigallocatechin and theaflavin increased SCE formation in CHO cells.

### 2.5. Micronuclei Formation

DNA damage during replication period can cause synthetic lethality. These DNA damages can be observed in the chromosome level as fragmented DNA (Figure 6A). Treatment of 10, 30, and 50 µM of tea polyphenol increased SCE frequency in a dose dependent manner for both V79 and CHO systems (Figure 6B,C). The regression lines showed micronuclei frequency increases of 0.0014 per µM of epigallocatechin treatment and 0.0009 per µM of theaflavin treatment for V79. The rate of increase was greater for V-C8 with epigallocatechin gallate or theaflavin treatment (0.039 and 0.006 per µM, respectively). The statistically significant difference of micronuclei formation was observed between V79 and V-C8 cells treated with 50 µM of epigallocatechin (*p* < 0.05). Compared to CHO cells, 51D1 also increased more MN formation with polyphenol treatment. Therefore, both epigallocatechin and theaflavin increased micronuclei formation to homologous recombination deficient cells.

### 2.6. DPPH Radical Scavenging Capacity

Radical scavenging capacity of two polyphenols were compared for DPPH (2,2-diphenyl-1-picrylhydrazyl) radical scavenging capacity (Figure 7). The DPPH radical scavenging capacity was much larger for epigallocatechin gallate than theaflavin. At the concentrations of 10 and 100 µM, statistically significant differences were observed (*p* < 0.05).

## 3. Discussion

In this study, we showed tea polyphenols can cause synthetic lethality in *BRCA2* deficient cells through PARP inhibition. In treated cells, cell doubling time was extended because tea polyphenol likely interferes with cell cycle progression and can postpone cell growth. This result is consistent with previous studies, which have shown tea polyphenols to induce perturbation of cell cycle progression in human prostate cancer cells [15]. Although, in this study the growth delay was not *BRCA2* deficient dependent (Figure 3), the clonogenic assay showed that a stronger cytotoxicity was induced by epigallocatechin gallate and theaflavin in *BRAC2* deficient cells (Figure 2). However, due to relatively weak specificity of PARP inhibition, the differences of IC_50_ for tea polyphenols induced cytotoxicity in wild type *BRCA2* cells and *BRCA2* deficient cells were not as large as known selective clinically used PARP inhibitors, such as olaparib [16]. These types of molecules found in tea are believed to still be healthy and safe to both types of normal and mutant *BRCA2*. While it is correct that the inhibition of PARP results in an inhibition of BER, however, cells that are *BRCA2* proficient (HR proficient), are able to repair double stranded breaks resulting from the accumulation of single stranded breaks. Moreover, cells that carry the *BRCA2* mutation are selectively killed, thus possibly preventing the formation of tumors specific to *BRCA2* status such as breast cancer and ovarian cancer [17].

In addition to PARP, another potential target of tea polyphenols are reported to be DNA topoisomerase II [18], histone deacetylases [19], and transcription factors [20]. This may explain why theaflavin displayed non-selective cytotoxicity to rad51D mutated cells even though they had defects in homologous recombination repair. Sister chromatid exchange was used for replicative stress analysis induced by PARP inhibitory effects by polyphenols (Figure 6). However, any genotoxic effects that directly or indirectly cause stress to cells can increase sister chromatid exchanges [21,22]. Therefore, observed elevation of sister chromatid exchange rate by tea polyphenols may not be solely from PARP inhibition. Another target of tea polyphenols are reported to be DNA topoisomerase II [18].

The PARP inhibition in vitro was stronger in theaflavin compared to epigallocatechin gallate. This was in clear contradiction with cell survival analysis and the two cytogenetics assays. Epigallocatechin gallate is more cytotoxic and genotoxic to CHO and V79 cells and their mutant cell lines. Epigallocatechin gallate is an ester of epigallocatechin (gallocatechol) and gallic acid. We have previously reported gallic acid as a PARP inhibitor [23]. Identified PARP inhibitory effect of epigallocatechin gallate supports that the galloyl group structure may be important for PARP inhibitory effects and synthetic lethality. This galloyl group structure is also reported to have an association with antioxidant capacity (Figure 7) [24]. Theaflavin itself does not have this structure. However, theaflavin-3-gallate is a theaflavin derivative. This chemical is found in black tea alongside theaflavin. Although both green tea and black tea contain polyphenols with PARP inhibitory effects, the metabolism and bioavailability of these polyphenols are not fully understood. The other natural chemicals that contain galloyl structure are epicatechin, epigallocatechin, catechin gallate, epicatechin gallate, gallocatechin, and gallocatechin gallate. Ethyl gallate may also possess PARP inhibitory effects. Myricetin, a flavonoid with a galloyl structure, may also possess PARP inhibitory effects because quercetin, a flavonoid with a catechol structure, shows PARP inhibitory effects [25].

Our study suggests that PARP inhibitory polyphenols in tea are not limited to epigallocatechin gallate and theaflavin. Further research could yield interesting results in the testing and analysis of other polyphenols in tea to induce PARP inhibition. Moreover, the PARP inhibition induced synthetic lethality was only tested through individual chemicals in this study. It is reasonable to speculate that polyphenols may show a synergistic effect for PARP inhibition and improvement of bioavailability due to forming complexes. Further research is necessary to assess the true tea polyphenols’ effects on PARP inhibition induced synthetic lethality in *BRCA2* deficient cells.

## 4. Materials and Methods

### 4.1. Cell Culture

Chinese hamster lung origin V79 and its DNA repair deficient V-C8 (BRCA2 deficient) [26], and their gene complimented V-C8 cells, Chinese hamster ovary origin CHO10B2 cells, were kindly supplied by Dr. Joel Bedford of Colorado State University (Fort Collins, CO, USA). CHO DNA repair deficient 51D1 (rad51D deficient) cells [27] were kindly supplied by Dr. Larry Thompson at the Lawrence Livermore National Laboratory (Livermore, CA, USA). Cells were maintained in Alpha MEM (Hyclone, ThermoFisher, Waltham, MA, USA) with 10% heat inactivated Fetal Bovine Serum (Sigma, St. Louis, MO, USA), antibiotics (Anti-Anti; Invitrogen, Grand Island, NY, USA), and were cultured in 37 °C incubators with 5% CO_2_ and humidity.

### 4.2. Chemicals

Green extract epigallocatechin gallate and black tea extract theaflavin were purchased from Sigma Aldrich (St Louis, MO, USA). These polyphenols were dissolved in DMSO as 10 mM or 100 mM stock solution and stored in −20 °C until experiments.

### 4.3. Clonogenic Cell Survival

Cells were plated in 6 well plates to form colonies. Various concentrations of tea extract were added to cell culture media and cells were allowed to incubate for 1 week. Following incubation, colonies were fixed and stained using 100% ethanol followed by 0.1% crystal violet. Macroscopic colonies containing more than 50 cells were considered as a survivor [10]. Regression curves were drawn from cell survival fraction by Graphpad Prism 6 software (GraphPad, La Jolla, CA, USA). IC_50_ values (drug concentration to achieve 50% cell survival) were obtained from regression curves.

### 4.4. Growth Inhibition

A quantity of 10,000 cells were plated onto each well of 12 well plates with 10 µM of chemicals and cell number was counted by Coulter Counter Z1 (Beckman Coulter, Indianapolis, IN, USA) every 24 h for 4 days. Cell doubling time was calculated by using GraphPad Prism 6 with exponential growing equation from exponentially growing stage.

### 4.5. In Vitro and In Vivo PARP Activity Inhibition Assay

PARP colorimetric assay kits (Trevigen, Gaithersburg, MD, USA) were used to measure in vitro PARP activity as previously described in test tubes [23]. PARP was incubated in a 96-well microplate with a reaction mixture containing 50 µM β-NAD+ (10% biotinylated β-NAD+), 90% unlabeled β-NAD+, 1 mM 1,4-dithiothreitol, and 1.25 mg/mL nicked DNA. The formation of poly (ADP-ribose) polymers was detected with peroxidase-labeled streptavidin and 3,3′,5,5′-tetramethylbenzidine (Invitrogen, Carlsbad, CA, USA). PARP inhibition was assessed by the addition of flavonoids at various dosages to the reaction mixture. PARP activity is directly proportional to absorbance at 450 nm and measured by a NanoDrop spectrophotometer (Thermo Fischer Scientific, Waltham, MA, USA). IC_50_ values of in vitro PARP inhibitory activity were derived by fitting dose–response curves using a sigmoidal dose response equation.

In vivo PARP activity inhibition was carried out as previously described in the cell culture system [23]. V79 were treated with polyphenols for 30 min before exposure to 2 mM hydrogen peroxide for minutes. After hydrogen peroxide treatment, cells were washed with PBS and fixed in 4% paraformaldehyde for 15 min, followed by 0.2% Triton X-100 in PBS for 10 min. After blocking with 10% goat serum in PBS, anti-poly (ADP-ribose) mouse monoclonal antibody diluted in 10% goat serum was added to the cells and incubated for 1 h at 37 °C. Alexa 488 conjugated anti-mouse antibody was used for the secondary antibody for fluorescence signal detection. After DNA staining with DAPI in slowfade (Thermo Fisher Scientific, Waltham, MA, USA), fluorescence images were obtained by a Zeiss Axiophot fluorescent microscope equipped with a Q-imaging Exi Aqua cooled CCD monochrome camera with Q-capture Pro software (Q-imaging, Surrey, BC, Canada). Green pixels per cell were scored for a minimum of 30 cells.

### 4.6. Sister Chromatid Exchange

CHO cells were synchronized into the G1 phase using a mitotic shake-off procedure [28,29]. Synchronized mitotic cells were subcultured in 6 well plates and incubated for two hours at 37 °C. Cells were then treated with various concentrations of testing chemicals and incubated with 10 µM of BrdU (Sigma, St Louis, MO, USA) for two cell cycles. Then 0.2 µg/mL of colcemid (Gibco, Invitrogen, Grand Island, NY, USA) was added to cells and allowed to incubate for an additional 6 h. Cells were harvested during metaphase, trypsinized, and then suspended in 2 mL of 75 mM KCl solution warmed to 37 °C and placed in a 37 °C water bath for 20 min. A fixative solution of 3:1 methanol to acetic acid was added to the samples according to the standard protocol [30]. Fixed cells were dropped onto slides and allowed to dry at room temperature. Differential staining of metaphase chromosomes was completed using the fluorescence plus Giemsa technique with Hoechst 33258 dye [31]. Differentially stained metaphase chromosome images were scored under a Zeiss Axioskop microscope equipped with a SPOT CCD camera RT 2.3.1 (Diagnostic Instrument, Inc., Sterling Heights, MI, USA) and SPOT basic software. A minimum of 50 metaphase cells were scored for each treatment concentration. Data presented are the mean of SCE frequency per chromosome.

### 4.7. Micronuclei Formation Assay

Cells were synchronized into the G1 phase by the mitotic shake off method [32]. Two h after shake off, testing drugs were added to the cell culture with 4 µg/mL of Cytochalasin B (Sigma, St Louis, MO, USA) for 22 h [33]. Harvested cells were suspended in 2 mL of 75 mM KCl solution and fixed in 1 mL of 3:1 methanol:acetic acid solution. Cells were dropped onto slides and allowed to air dry at room temperature. Slides were stained in 5% Giemsa solution in Gurr solution (Invitrogen) for 5 min. Then, 100 binucleated cells were scored per data points under Zeiss Axioskop microscope to obtain micronuclei frequency per binucleated cells.

### 4.8. DPPH Analysis

DPPH analysis was performed as previously described [34]. First, 100 µL of 100 μM DPPH in ethanol, 80 µL of ethanol, and 20 µL of tea polyphenol were mixed. The mixture was agitated vigorously and allowed to stand at room temperature for 30 min. Absorbance was measure at 490 nm using a microplate reader. DPPH scavenging activity was calculated by absorbance with samples divided by absorbance of control. All experiments were carried out at least three times and error bars indicate standard error of the means.

### 4.9. Statistics

All experiments were carried out at least three times and error bars indicate standard error of the means. Data was analyzed using Prism 6 software for one-way ANOVA analysis; *p*-values < 0.05 were categorized as significant differences.

## 5. Conclusions

In conclusion, this paper identified tea polyphenols, especially epigallocatechin gallate and theaflavin, to have PARP inhibitory activity and selective cell toxicity towards BRCA2 deficient cells through synthetic lethality. This study also supported that the galloly group structure of epigallocatechin gallate is important for PARP inhibition. Further research should be done to investigate in vivo synthetic lethality cancer preventative activities and the PARP inhibitory mechanisms of these polyphenols.

## Figures and Tables

**Figure 1 ijms-20-01274-f001:**
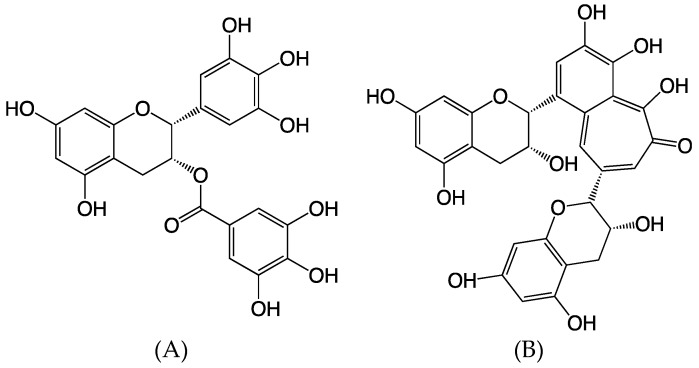
Chemical structures of tea polyphenols. (**A**) Epigallocatechin gallate. (**B**) Theaflavin.

**Figure 2 ijms-20-01274-f002:**
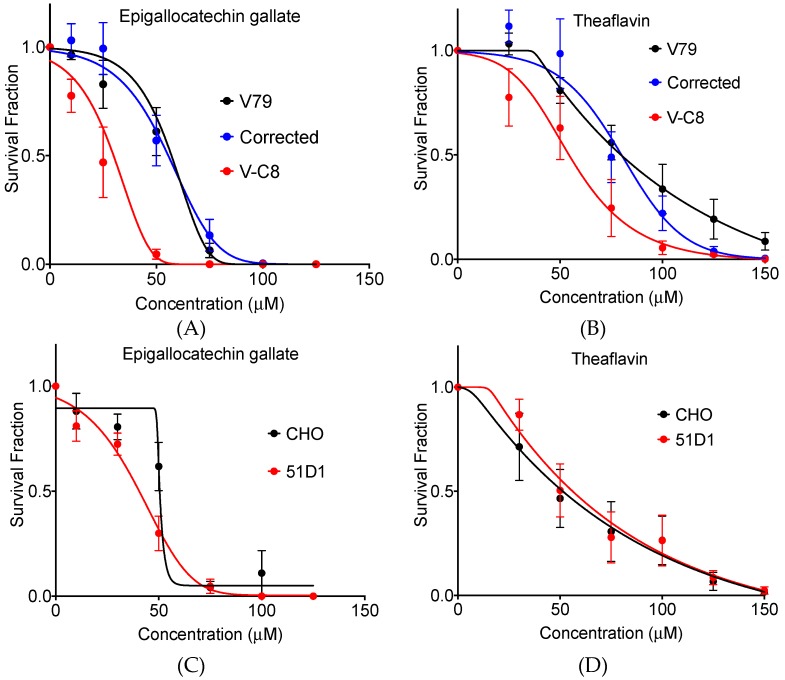
Clonogenic cell survival curves against tea polyphenol epigallocatechin gallate and theaflavin. (**A**) Epigallocatechin gallate toxicity to V79 cells, V-C8 cells, and V-C8 gene complimented cells. (**B**) Theaflavin toxicity to V79 cells, V-C8 cells, and V-C8 gene complimented cells. (**C**) epigallocatechin gallate toxicity to Chinese hamster ovary (CHO) wild type cells and 51D1 cells. (**D**) Theaflavin toxicity to CHO wild type cells and 51D1 cells. Error bars represent standard error of the means. At least three independent experiments were carried out.

**Figure 3 ijms-20-01274-f003:**
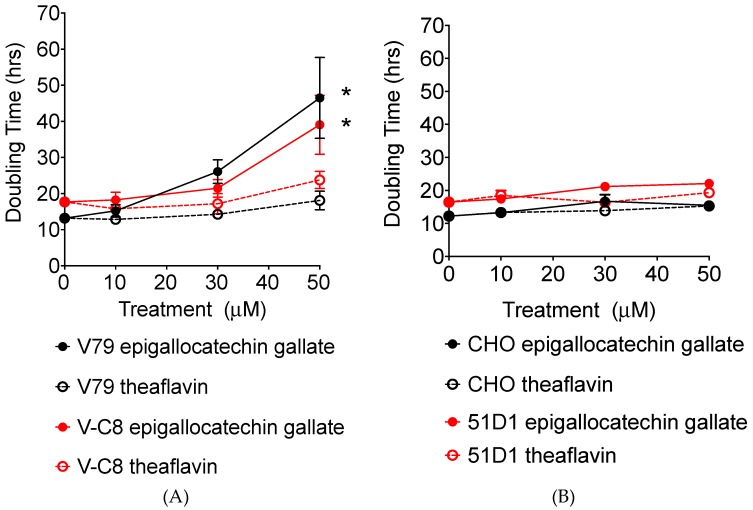
Cell population doubling time with polyphenol treatment. (**A**) V79 and V-C8 cells with polyphenols. (**B**) CHO and 51D1 with polyphenols. Values are mean ± standard error of the means. * indicates statistically significant differences from control (*p* < 0.05).

**Figure 4 ijms-20-01274-f004:**
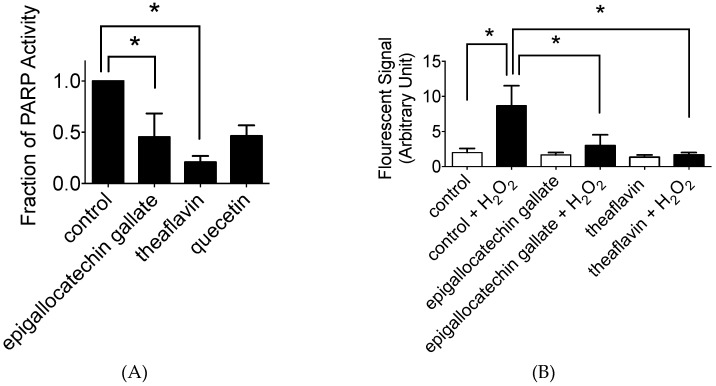
PARP inhibitory effect by tea polyphenols. (**A**) 10 µM of chemicals were tested for in vitro analysis. (**B**) 10 µM of chemicals were tested for in vivo analysis. V79 cells were pre-treated with polyphenol prior to H_2_O_2_ treatment. Values are mean ± standard error of the means. * indicates statistically significant differences from control (*p* < 0.05).

**Figure 5 ijms-20-01274-f005:**
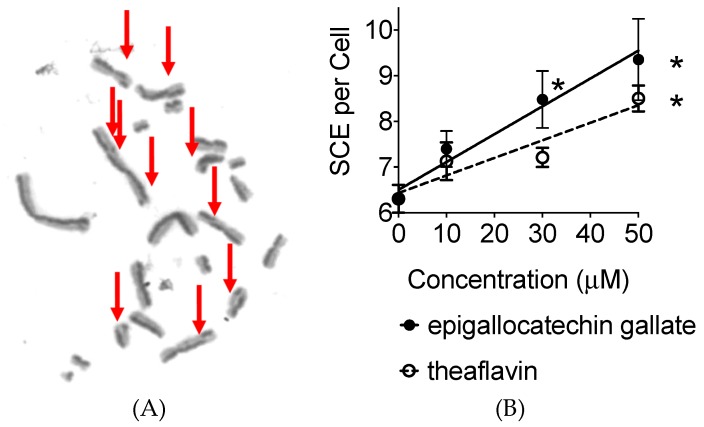
Sister chromatid exchange (SCE) induction by polyphenol treatment. (**A**) SCE after 50 µM of epigallocatechin gallate treatment. Arrows indicate SCE formation. (**B**) The dose response increases in sister chromatid exchange frequency with the treatment of polyphenols. Linear regression lines were drawn. Epigallocatechin gallate SCE = 0.061 × (epigallocatechin gallate µM) + 6.5, theaflavin SCE = 0.038 × (theaflavin µM) + 6.4. Values are mean ± standard error of the means. * indicates statistically significant differences compared to 0 μM control (*p* < 0.05).

**Figure 6 ijms-20-01274-f006:**
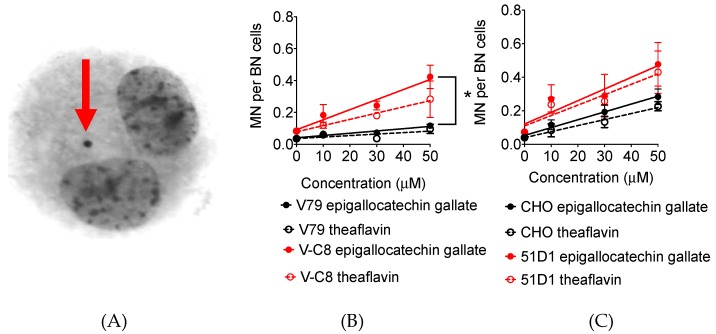
Micronuclei formation by tea polyphenol treatment. (**A**) Micronuclei formation after epigallocatechin gallate 50 µM treated 51D1 cells. Arrow indicates micronuclei in binucleated cells. (**B**) Dose dependent micronuclei formation for V79 and V-C8 cells. (**C**) Dose dependent micronuclei formation for CHO and 51D1 cells. Values are mean ± standard error of the means. * indicates statistically significant differences from non-treated data (*p* < 0.05).

**Figure 7 ijms-20-01274-f007:**
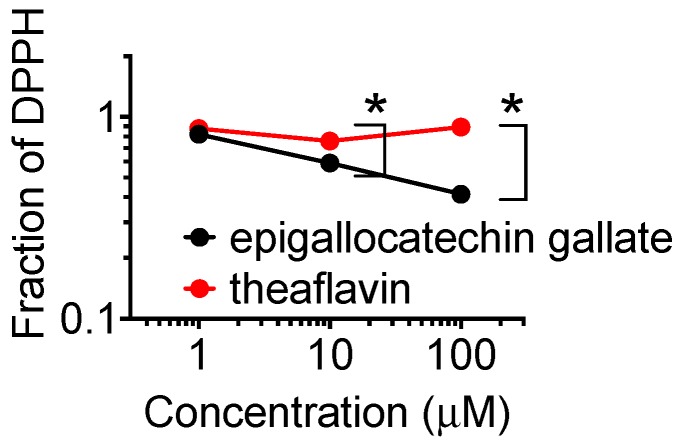
DPPH radical scavenging capacity of two tea polyphenols. Data represents means and standard error of the means. * indicates statistically significant differences from non-treated data (*p* < 0.05).

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
