# Peer review of "The Effect of Green and Black Tea Polyphenols on BRCA2 Deficient Chinese Hamster Cells by Synthetic Lethality through PARP Inhibition"

_ijms, 2019, doi:10.3390/ijms20061274_

Round 1

Reviewer 1 Report

This manuscript is investgated that the cytotoxicity of epigallocatechin gallate and theaflavin was shown to BRCA2 deficient cells through synthetic lethality induced by PARP inhibition.

There are some corrections as follows:

・L71: damage with → damage with

・L104: Please add   (Figure 2C and 2D)

・L188: (Figure 6B) → (Figure 6B and 6C)

・L326: wrote the paper."→wrote the paper.

Author Response

Thank you very much for reviewing our manuscript. Our answer is attached.

Reviewer 2 Report

Two tea polyphenols were investigated for their PARP inhibitory activity and selective cytotoxicity to BRCA2 mutated cells. And the observed cytotoxicity of these polyphenols is believed to be a result of PARP inhibition induced synthetic lethality. Several BRCA2 deficient but also complemented cell lines were investigated. The epigallocatechin gallate polyphenol showed cytotoxicity to V-C8 cells and rad51D mutant cells less to complemented cells. Theaflavin showed cytotoxicity effect to VC8 cells but not to rad51D mutant. All compared to wt cells. It was concluded that tea polyphenols showed selective cytotoxicity to BRCA2 deficient cells through synthetic lethality induced by PARP inhibition.

 Is tea now unhealthy for people with or without a BRCA2 mutation? I presume when parp is inhibited, BER is not functioning and as in a human body in the DNA hundreds of thousands DNA damages occur inhibition of BER is no good!

Or is it  good for people with cancer with a BRCA2 mutation? So, would patients with a BRCA2 mutation induced tumor benefit from drinking tea? Usually people with mutation have one allele mutated and in the tumor both alleles are mutated. I thought that especially green tea is healthy.

How come the polyphenols are active on BRCA2 mutated cells but not rad51 mutant cells? Homologous recombination is impaired in both cell types.  

What is the concentration in tea of these polyphenols compared to the concentrations used here?

What is the sensitizing effect of the polyphenols to ionizing radiation and/or cisplatin?

What is exactly meant with in vitro and in vivo as both are in cell culture systems.

The observed elevation of sister chromatid exchange rate by tea polyphenols may not be solely from PARP inhibition. Could the authors speculate what other mechanisms might be involved?

In line 19: V-C8 with gene ‘complimented’ cells were tested,  should be:  ‘complemented’.

Author Response

Thank you very much for reviewing our manuscript. Our answers are attached.
